# Effect of Different Anchorage Reinforcement Methods on Long-Term Maxillary Whole Arch Distalization with Clear Aligner: A 4D Finite Element Study with Staging Simulation

**DOI:** 10.3390/bioengineering11010003

**Published:** 2023-12-20

**Authors:** Bochun Mao, Yajing Tian, Yujia Xiao, Jiayi Liu, Dawei Liu, Yanheng Zhou, Jing Li

**Affiliations:** 1Department of Orthodontics, Peking University School and Hospital of Stomatology, National Center for Stomatology, National Clinical Research Center for Oral Diseases, National Engineering Research Center of Oral Biomaterials and Digital Medical Devices, Beijing Key Laboratory of Digital Stomatology, Research Center of Engineering and Technology for Computerized Dentistry Ministry of Health, Beijing 100081, China; heyyoobc@bjmu.edu.cn (B.M.); 1398743593@pku.edu.cn (Y.X.); 1810303110@pku.edu.cn (J.L.); liudawei@bjmu.edu.cn (D.L.); yanhengzhou@vip.163.com (Y.Z.); 2Center of Stomatology, China–Japan Friendship Hospital, Beijing 100013, China; 1510303118@pku.edu.cn

**Keywords:** orthodontics, finite element method, long-term simulation, clear aligner, molar distalization, tooth movement

## Abstract

The objective of this study was to examine how various anchorage methods impact long-term maxillary whole arch distalization using clear aligners (CAs) through an automated staging simulation. Three different anchorage reinforcement methods, namely, Class II elastics, buccal temporary anchorage device (TAD), and palatal TAD, were designed. Orthodontic tooth movement induced by orthodontic forces was simulated using an iterative computation method. Additionally, the automatic adjustment of the CA was simulated through the application of the thermal expansion method. The results indicated that the palatal TAD group had the largest retraction of incisors, followed by the buccal TAD group and the Class II elastic group, while the least was in the control group. The largest distal displacements and efficiency of molar distalization for the first and the second molars were noticed in the palatal TAD group. Arch width increased at the molar and premolar levels in all groups. The FEM results suggested palatal TAD had the best performance considering anterior teeth anchorage maintenance, both sagittally and vertically. However, attention should be paid to the possible increasement of arch width.

## 1. Introduction

In the past two decades, clear aligners (CAs), a new orthodontic appliance, have been widely used due to their advantages of comfort, hygiene, and esthetics [1,2,3,4]. In the context of CA treatment, it is noteworthy that molar distalization exhibits a high expression rate of 88% compared to other designed tooth movements [5]. Consequently, molar distalization stands as a favored treatment approach for patients seeking to attain a 2–3 mm arch space and establish a Class I molar relationship. Nonetheless, during molar distalization, the reacting forces may lead to the proclination of anterior teeth. Therefore, the dental space found during subsequent visits may be partly due to the mesial movement of the anterior teeth, which disturbs clinical decision making [6]. Furthermore, previous clinical research has revealed that in the course of total arch distalization with CAs, the molars that have already undergone distalization may experience mesial movement in subsequent stages, ultimately compromising the effectiveness of the intended movement [7]. Various methods for anchorage reinforcement have been employed in clinical practice, including interarch elastics, buccal interradicular temporary anchorage devices (TADs), palatal TADs, and the like. However, the elastics from different angles may cause problems such as the poor retention of CAs or intrusion or extrusion of anterior teeth. Consequently, the most optimal method for anchorage reinforcement in maxillary total arch distalization remains indeterminate.

With a profound comprehension of the biomechanics of CAs, orthodontists may attain treatment outcomes that are not only secure, but also foreseeable and enduring. The finite element method (FEM), an analytical technique enabling the examination of stress distribution and distortion within a given structure, serves as a valuable tool for investigating the intricate biomechanical dynamics of CAs. Nonetheless, it is important to note that all the studies published thus far have been confined to examining the initial displacement occurring during CA wear. This limitation underscores the need for more comprehensive investigations to yield results that can serve as more robust clinical references. The forces and movements change constantly with changes in the position and contact of the CA with the tooth crown, which further weakens the clinical significance of the results of conventional FEM studies.

In the realm of CAs, the prediction and staging of orthodontic tooth movement have predominantly relied upon clinical expertise. In our earlier research, we introduced the concept of the ‘four-dimensional (4D)’ FEM [8]. This innovative approach integrates biomechanical responses as an additional dimension, enabling long-term orthodontic simulations of CA treatment. However, only 20 steps of distalization of the second molar were simulated, without any comparison of distinct anchorage designs.

Hence, the primary objective of this study was to examine the impact of various anchorage designs on long-term maxillary whole arch distalization using clear aligners. We employed an innovative automatic staging simulation method previously introduced in one of our earlier studies [8].

## 2. Method

This retrospective study received ethical approval from the bioethics committee at the Peking University School and Hospital of Stomatology (No. PKUSSIRB-202059154). The FEM model employed in this study included teeth, the periodontal ligament (PDL), attachments, and CAs, as illustrated in Figure 1a. The construction of this model followed the established methodology described in previous publications [9,10,11]. Beside the control group, three test groups with different anchorage designs were carried out (Figure 1b): (1) buccal TAD group: the TAD was located in the buccal interradicular space between the first and second molars 4 mm above the alveolar crest; (2) palatal TAD group: the TAD was located in the palatal interradicular space between the first and second molars 4 mm above the alveolar crest; (3) Class II elastic group: the extraction force was distal pointed, 15° below the occlusal plain. The application of elastics on the CAs were all set at the buccal or lingual mesial cervical region of the canine to simulate the precision cut. According to previous studies and clinical experiences [12,13], the elastic forces for all the test groups were set as 150 g in this study.

In regard to the CAs and attachments, the elastic modulus values were set at 1500 MPa and 20,000 MPa, respectively, with Poisson’s ratios of 0.30 for both [14]. As for the PDL, the nonlinear hyperelastic model based on the double linear stress–strain curve was used [15]. The assembly of all models were accomplished using Hypermesh 14.0 software (Altair, Troy, MI, USA). Unstructured four-noded tetrahedral elements were employed for the meshing process. Subsequently, the models were imported into Abaqus (SIMULIA, Providence, RI, USA). Positional constraints were imposed on the PDLs and tooth roots. The interaction between the aligners and crowns was defined as a small-sliding surface-to-surface contact, with a friction coefficient set at 0.2 [16]. To ensure accuracy, a convergence study was conducted to determine the optimal element size. The results indicated that stress and displacement converged when the element size was smaller than 0.2 mm.

The bone remodeling process during CA treatment was simulated, as disclosed previously [17]. This two-step calculation process was iterated to simulate the long-term orthodontic tooth movement, as illustrated in Figure 2.

The occlusal plane was delineated based on specific reference points, namely, the mesial-buccal cusps of the upper first molars and the central incisors’ midpoint. The origin for the coordinates was set at the central incisors’ midpoint. The X-axis was established parallel to the line connecting the mesial-buccal cusps of the upper first molars, while the Y-axis, perpendicular to the X-axis, was also defined, as illustrated in Figure 3a. Under the occlusion view, the center point of each dental crown was determined (Figure 3b) and recorded to draw the arch form with the polynomial regression algorithm in Matlab, and fourth-order polynomials were selected for curve fitting according to a previous study [18] (Figure 3c). During staging, all the teeth except the incisors move along the fitting curve, and the incisors move palatally. The common ‘V pattern’ staging strategy for maxillary whole arch distalization was designed (Figure 4). During the total 70 steps, for the movement of the canine to the second molar, 0.1 mm distal movement along the arch form was prescribed for the target teeth; for the movement of the incisors, 0.15 mm palatal movement was designed in each relevant step.

As previously described, the temperature changing method (TCM) was employed in this study for the automatic remodeling of the CAs during long-term orthodontic simulations [8]. As shown in Figure 4, the process began with the determination of the center point of dental crowns in the occlusal view, denoted as (C_i_, C_j_). Margin points of dental crowns on the line connecting C_i_ and C_j_ were established as (P_i_, P_j_), and the center point of P_i_–P_j_ (P_c_) was then calculated. A 1 mm area, both mesial and distal to P_c_, perpendicular to the C_i_–C_j_ line, was identified as the deformation region during staging. Deformation within this area was constrained along the C_i_–C_j_ line. The deformation within this region is governed by the following formulas:(1)U=k⋅(d+∑Δ)⋅t
(2)$${\left(X,Y\right)}_{N_{i}}(n)={\left(X,Y\right)}_{N_{i}}(n-1)+U\cdot \overset{\to }{e}\cdot \lambda$$

In Formula (1), U refers to the preset deformation quantity, k refers to the coefficient of linear expansion, Δ refers to the amount of deformation in previous steps, d refers to the width of the area, and t refers to the change of temperature. In Formula (2), (X,Y) refers to the coordinate value of Ni (Figure 3b,c), n refers to the numbers of moving steps of the target tooth, $\overset{\to }{e}$ refers to the component vector of the demand deformation direction, and λ refers to the displacement control factor. By establishing the specified values, the CA within the interdental area can autonomously undergo precise adjustments, eliminating the necessity for manual remeshing, as illustrated in Figure 5. Appendix A shows the automatic morphological changes of CAs during the 70 steps.

A preliminary experiment suggested that for each CA, less than 0.1% strain of PDL was noticed during the third iteration of PDL, which proved that the model remained steady after the first two iterations of PDL. Therefore, during the staging of CA treatment, each step of the CA was accompanied by two iterations of PDL to simulate the clinical situation that the CA was well worn with enough time. Prior to each iteration, the CA from the previous step was removed and regenerated, followed by the application of a best-fit algorithm to match the inner surface of the CA with the dental crowns, simulating a wear-in process. These operations were automated through custom Python subroutines designed for ABAQUS. Prior to each iteration, the CA from the previous step was removed and regenerated, followed by the application of a best-fit algorithm to match the inner surface of the CA with the dental crowns, simulating a wear-in process. These operations were automated through custom Python subroutines designed for ABAQUS.

The crown point, the center of resistance point (CR), and the long axis (LA) were defined according to one of our previous studies [8]. Throughout the simulation, the 3D displacements of the crown points and the rotations of the LA were continuously recorded at each step.

## 3. Results

Long-term maxillary whole arch distalization by CAs with different anchorage designs was achieved. Appendix A shows the movements of teeth during the total arch distalization staging process. Detailed quantified results are shown in Appendix A.

In the initial 10 steps of all groups (Figure 6, Figure 7 and Figure 8), with the distal tilting of the second molar, there was a noticeable flaring of the incisors and canines, which was particularly prominent in the control group (1.03–1.21°). The control group exhibited the least distal movement and tilting of the second molar (0.97 mm, 5.92°) compared to the other groups (1.09–1.13 mm, 6.63–6.87°). However, the movement tendencies of the second molars of all groups were similar, with the center of the rotation of the located around the apical third (Figure 9). Theoretically, the largest anchorage requirement occurred in steps 10–20, during which both of the two molars were prescribed with distal movements. Indeed, obvious mesial controlled tipping of the premolars, uncontrolled mesial tipping of the canines, and uncontrolled labial tipping of the incisors were noticed during steps 10–20, with the most pronounced coronal axis rotation of the incisors displaced in the control group (3.63–4.05°) and the least rotation exhibited in the palatal TAD group (2.38–2.55°). In the initial 20 steps, all the distal movements of the first or second molars in all groups were recognized as uncontrolled tipping, and the displacement tendencies were similar (Figure 9). In steps 20–40 across all groups, undesired mesial tipping was observed in the already distally shifted second molars, with the greatest displacement occurring in the control group (0.89 mm) and movements ranging from 0.61 to 0.76 mm in the other groups. The largest uncontrolled labial tipping of the incisors was exhibited at step 50 in all groups. The palatal TAD group demonstrated the least labial inclination, accompanied by central incisor intrusion (6°, 1.21 mm), in contrast to the other groups (9.27–10.26°, 1.87–2.14 mm). During steps 50–70, incisors of all groups showed controlled palatal tipping.

As for the dentition after the whole distalization process (at the 70th step, Figure 10), the palatal TAD group exhibited the most significant retraction of the incisors (1.88 mm), followed by the buccal TAD group (1.71 mm) and the Class II elastic group (1.16 mm), while the control group displayed the least retraction (0.87 mm). Despite sagittal movements, for the central incisor, 0.3 mm extrusion along with 6° lingual inclination was found in the palatal TAD group; however, 1.80 mm, 1.08 mm, and 0.95 mm intrusion with 1.9°, 0.65°, and 0.50° labial inclination were found in the buccal TAD group, control group, and Class II elastic group, respectively. The palatal TAD group demonstrated the most substantial distal displacements and efficiency in molar distalization for both the first and second molars (1.00 mm, 51.55%; 1.31 mm, 67.53%), with the least of those in the control group (0.62 mm, 31.96%; 0.73 mm, 37.63%). The Class II elastic group exhibited more incisor extrusion and molar intrusion, attributed to the clockwise rotation of the occlusion plane compared to the buccal TAD group. Additionally, all groups experienced an increase in arch width at the molar and premolar levels. The greatest buccal displacement of the dental crown was observed in the first molar and second premolar, with the palatal TAD group displaying the largest displacement (1.17 mm, 1.14 mm), followed by the control group (0.85 mm, 0.82 mm), buccal TAD group (0.70 mm, 0.57 mm), and Class II elastic group (0.59 mm, 0.40 mm).

## 4. Discussion

In this study, long-term maxillary whole arch distalization with CAs was simulated for the first time and different anchorage designs were investigated. Our previous study [8], which introduced the TCM for CA staging simulation, solely focused on the linear expansion of CAs during the distalization of the second molar. However, this study extended the application of the TCM to the simulation of movement of teeth angulate with the arch curve during CA treatment.

In the realm of current biomechanical research based on the FEM, the majority of studies have been confined to examining the initial mechanical conditions and displacements within the periodontal ligament (PDL) and other models. Regrettably, these analyses failed to provide comprehensive insights into the long-term orthodontic tooth movement that occurs during orthodontic treatment. To address this limitation, several iterative computational methods aimed at simulating tooth movement throughout orthodontic treatment were proposed. However, all the previous methods have been exclusively applicable to a fixed appliance, which is easier to realize compared to CA treatment. In this study, we employed the method previously elucidated by Hamanaka et al. [17]. This approach eliminates the necessity of modeling the surrounding alveolar bone, resulting in significant time savings and enhanced convergence during multi-step simulations. As revealed in our previous study [8], the TCM was used in this study for CA morphological changing, which saw automatically precise CA changing realized.

As recommended by the manufacturer, ‘V pattern’ staging strategy is commonly used during molar distalization. It was believed that with ‘V pattern’ staging, anchorage could be best preserved [19]. The interradicular area and midpalatal area are the two main areas for palatal TAD setting [20,21,22]. However, most of the midpalatal TADs used for sagittal anchorage reinforcement were used together with support frames for traction [20,21], since the direct elastic traction which connects the aligner and the midpalatal TADs loosens easily and is clinical uncomfortable. With the support frames, the direction and site of elastic traction is similar to interradicular palatal TADs. Therefore, only one palatal TAD group was determined to simulate both the interradicular palatal TADs scenario and the midpalatal TADs with a support frame scenario, and another simulation group may not be needed theoretically. As shown in the results, the difference in the efficiency of molar distalization and the loss of anchorage among the control group and the other groups are obvious, which proved the necessity of anchorage reinforcement methods. Since a greater sagittal component of force exists with retractive forces from palatal TAD than other groups, the best results were revealed in the palatal TAD group, followed by the buccal TAD group and the Class II elastic group. Additionally, the use of Class II elastics can lead to side effects such as mesialization and extrusion of the mandibular molar, while lingual tipping and the extrusion of maxillary incisors can occur [23]. Measures to enhance anchorage should be implemented to prevent bone fenestration or bone dehiscence if thin labial side alveolar bone is observed. In accordance with our own findings, Ravera et al. reported similar proclination of incisors during molar distalization [24]. Similarly, another previous study noticed an average of 2–11° distal tilting with less than 2 mm intrusion of molars during the distalization movement design with CAs [25], especially for the second molars. Likewise, a previous study indicated that a different amount of traction forces within a certain range would not have a significant influence on teeth movement patterns during CA treatment [26]. During the first 20 steps, no significant difference was noticed with the molar distal tipping tendencies among the four groups. During the 70 steps, almost all the teeth movements were uncontrolled tipping, with only a few controlled tipping. This fact emphasizes the importance of overcorrection and the design of attachments during CA treatment.

Kwak et al. [27] conducted a comparison between direct (using a button on the canine) and indirect (precision cut on the CA) posterior retraction methods using the FEM during maxillary total distalization, and their findings indicated that the posterior movement pattern was significantly influenced by the presence or absence of attachments and the direction of the force, rather than the specific area where the force was applied. However, other previous studies suggested that during the molar distalization, different attachment designs had a limited impact on the efficacy of the designed movement [28].

In addition, during the whole arch distalization process, the previously distalized molars may exhibit mesial movement in subsequent steps, as previously attested by clinical research [7]. During the first 50 steps, in which labial inclination along with intrusion of incisors accumulated, a ‘reversed bow effect’ was obvious for all groups [8]. Therefore, retractive forces with positive vertical vectors (directed downward towards the occlusal plane) are conductive for resisting the potential intrusion of incisors, and other overcorrection design should be made. In this study, during the 70 steps of all groups, the largest gap happened at the second molar at the 20th step, which is less than 1.2 mm. No obvious offtrack was noticed during the simulation, which may be due to the wear-in process before the calculation. Vertically, unlike the two groups with TADs, Class II elastics offer more dislocation force than other traction groups, which in a clinical environment may lead to dislocation of CAs. From a horizontal perspective, palatal traction may lead to wider CAs at the posterior area, which ultimately lead to wider dentition. The direction of traction forces in the other two test groups all close to the long axis of dental arch buccally, which is a benefit for width stabilization. Previous studies [29] have reported that Invisalign Class II treatment with maxillary molar distalization significantly increased arch width at the molar and premolar levels, which was consistent with our results. Moreover, from a sagittal perspective, during molar distalization, the labial inclination of incisors leads to weak retention, which in turn decreases the efficacy of the posterior teeth distalization.

The results of our previous study showed that approximately 68% designed distal movement of molars were achieved without anchorage reinforcement [8], which was significantly smaller than the rate of 88% previously proposed by Simon et al. [30]. Furthermore, only 37.63% and 67.53% efficacy of the second molar distalization were found in the control and palatal TAD groups, respectively, which was in accordance with a previous retrospective study [31]. The finding of significant intrusion and buccal tipping of the second molar was also similar to our results. Possible clinical strategies to improve the expression rate of designed teeth movement include the design of over-correction and prolonging the wearing time of each CA.

The limitations of this study were also obvious, and further studies are needed. For the first time, this study proposed a feasible long-term orthodontic simulation method for CA treatment. However, it was only when combined with further in vitro experiments and clinical trials that the accuracy of the orthodontic tooth movement simulation method could be certified and improved. Also, in this study, elastics were all exerted directly on the CA, simulating the use of precision cuts. Traction forces from lingual buttons or power arms attached to the teeth should also be investigated in further studies. Furthermore, all test groups have the same traction force magnitude set as 150 g. However, different amount of traction forces may have different biomechanical consequences, due to the potential offtrack of the CA, which should be further investigated. Moreover, a lot of simplification was made in the current bone remodeling simulation method to gain a balance between the computational efficiency and accuracy, while its clinical accuracy should also be justified. However, we believe that this study opens a door for long-term orthodontic simulation for CA treatment. Despite the current proposed maxillary whole arch distalization scenario, with the development of the FEM, more accurate and clinically significant results for all kinds of teeth movement design can be gained in the future.

## 5. Conclusions

In this study, a time-dependent 4D FEM was developed to predict orthodontic tooth movement during maxillary dentition distalization with CAs. Using this method, the morphological changes of CAs during long-term orthodontic treatment were successfully simulated. The achieved results indicated:Compared with other anchorage reinforcement measurements such as Class II elastics and buccal TADs, palatal TADs were recommended during maxillary dentition distalization with CAs for better anterior teeth anchorage maintenance, both sagittally and vertically.Attention should be paid to the possible increasement of arch width at the molar and premolar levels.The overbite of front teeth may become shallow during the process, and timely compensation measures should be carried out.

## Figures and Tables

**Figure 1 bioengineering-11-00003-f001:**
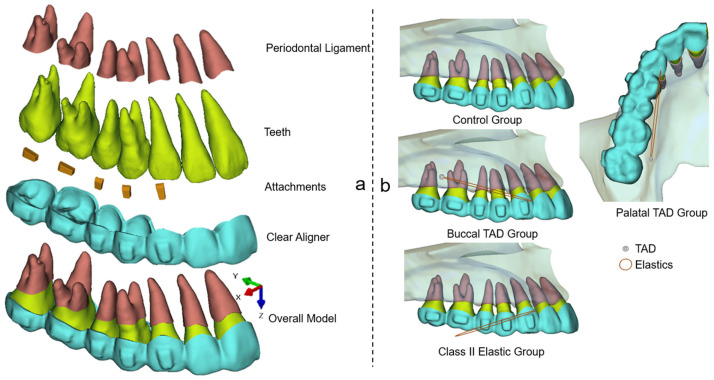
Components of the model (**a**) and the different anchorage design groups (**b**).

**Figure 2 bioengineering-11-00003-f002:**
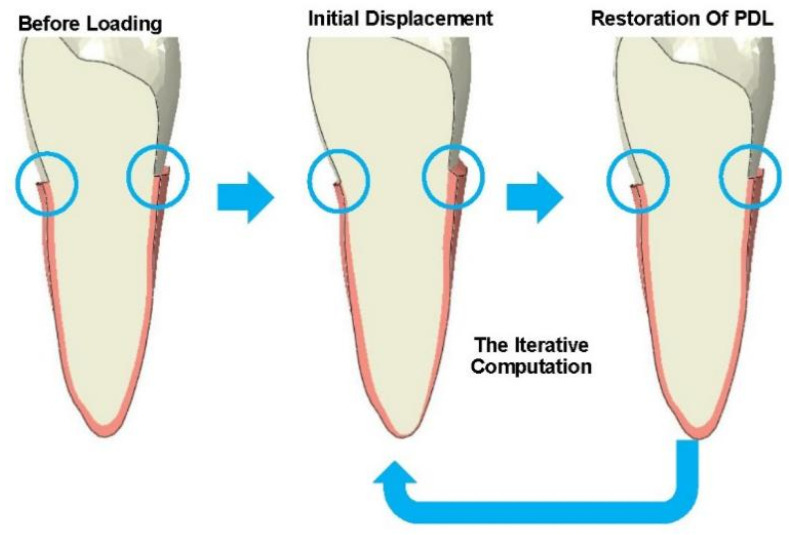
The bone remodeling simulation method, highlighting the deformation of the periodontal ligament, indicated within the blue circles. PDL, periodontal ligament.

**Figure 3 bioengineering-11-00003-f003:**
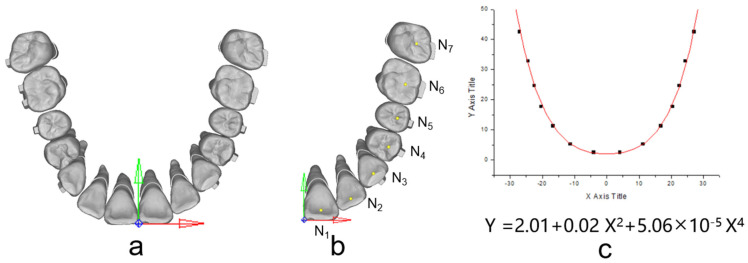
(**a**) The establishment of the coordinate system; (**b**) The determination of the center point of each dental crown (yellow dots); (**c**) Curve fitting with a fourth-order polynomial function. The tooth numbering system adheres to the FDI tooth numbering system.

**Figure 4 bioengineering-11-00003-f004:**
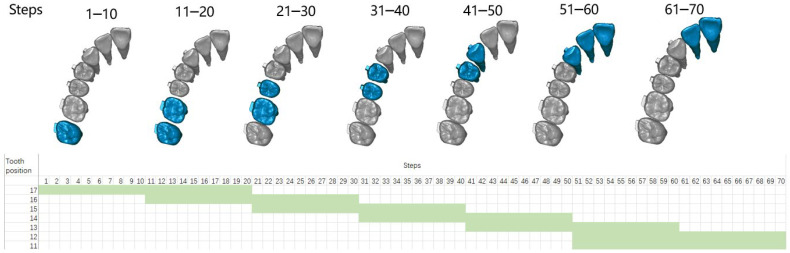
The ‘V pattern’ staging strategy was designed in this study: during the total 70 steps, the cells referred to designed movement were set green and the corresponding teeth were marked in blue. The tooth numbering system adheres to the FDI tooth numbering system.

**Figure 5 bioengineering-11-00003-f005:**
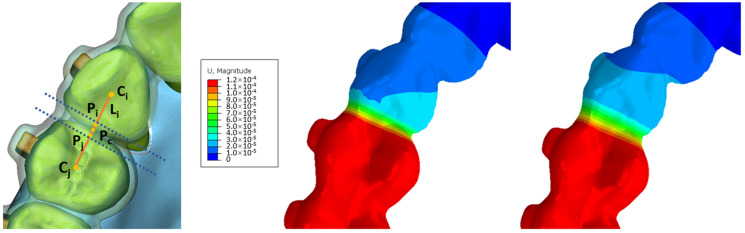
The automated clear aligner adjustment method relies on a temperature changing method. C_i_–C_j_, the central point of dental crowns; P_i_–P_j_, the boundary points of dental crowns situated along the C_i_–C_j_ line; Pc, the center point of P_i_–P_j_. L_i_, the line of C_i_ to C_j_.

**Figure 6 bioengineering-11-00003-f006:**
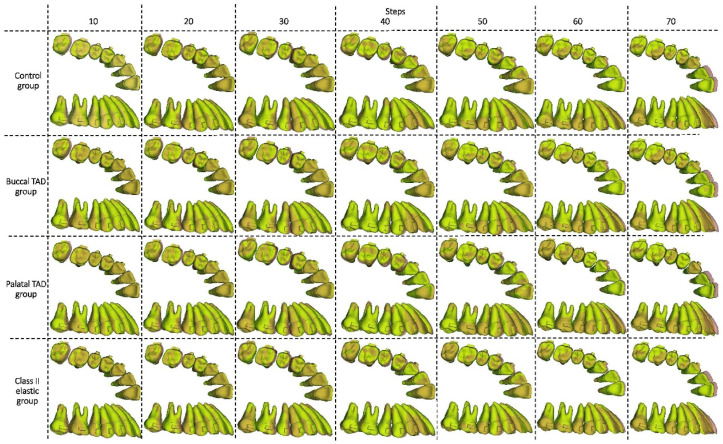
Teeth displacement at different treatment staging steps of the four groups. Each illustration presents the comparation of the nth steps (yellow) and the (*n* − 10)th steps (orange) of the four groups (*n* = 10, 20, 30, 40, 50, 60, 70). TAD, temporary anchorage device.

**Figure 7 bioengineering-11-00003-f007:**
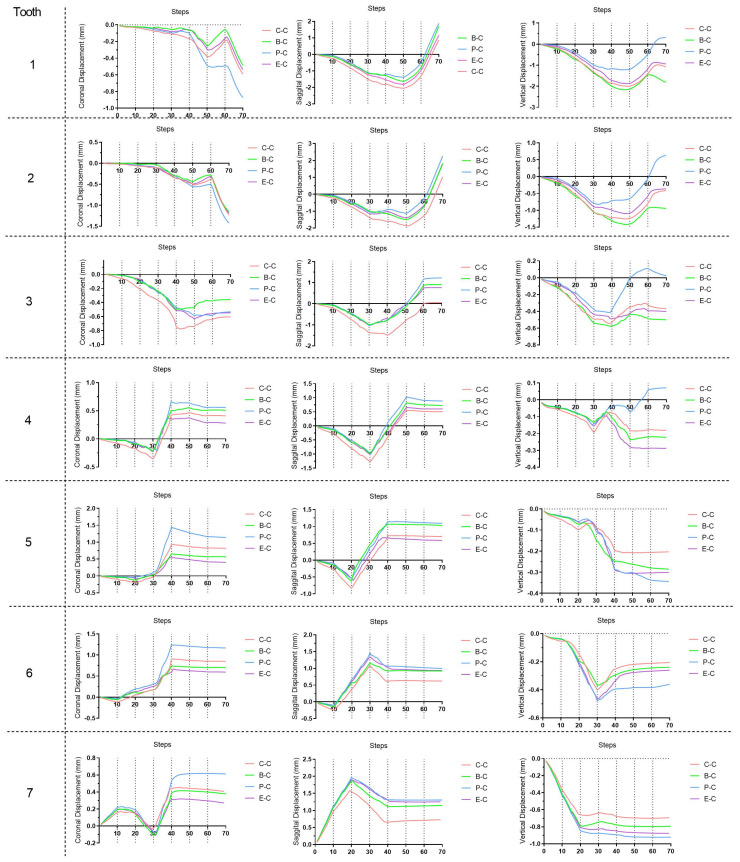
Three-dimensional displacement of the crown points of each tooth during the 70 steps. The tooth numbering system adheres to the FDI tooth numbering system. C–C, crown point of control group; B–C, crown point of buccal temporary anchorage device (TAD) group; P–C, crown point of palatal TAD group; E–C, crown point of Class II elastic group.

**Figure 8 bioengineering-11-00003-f008:**
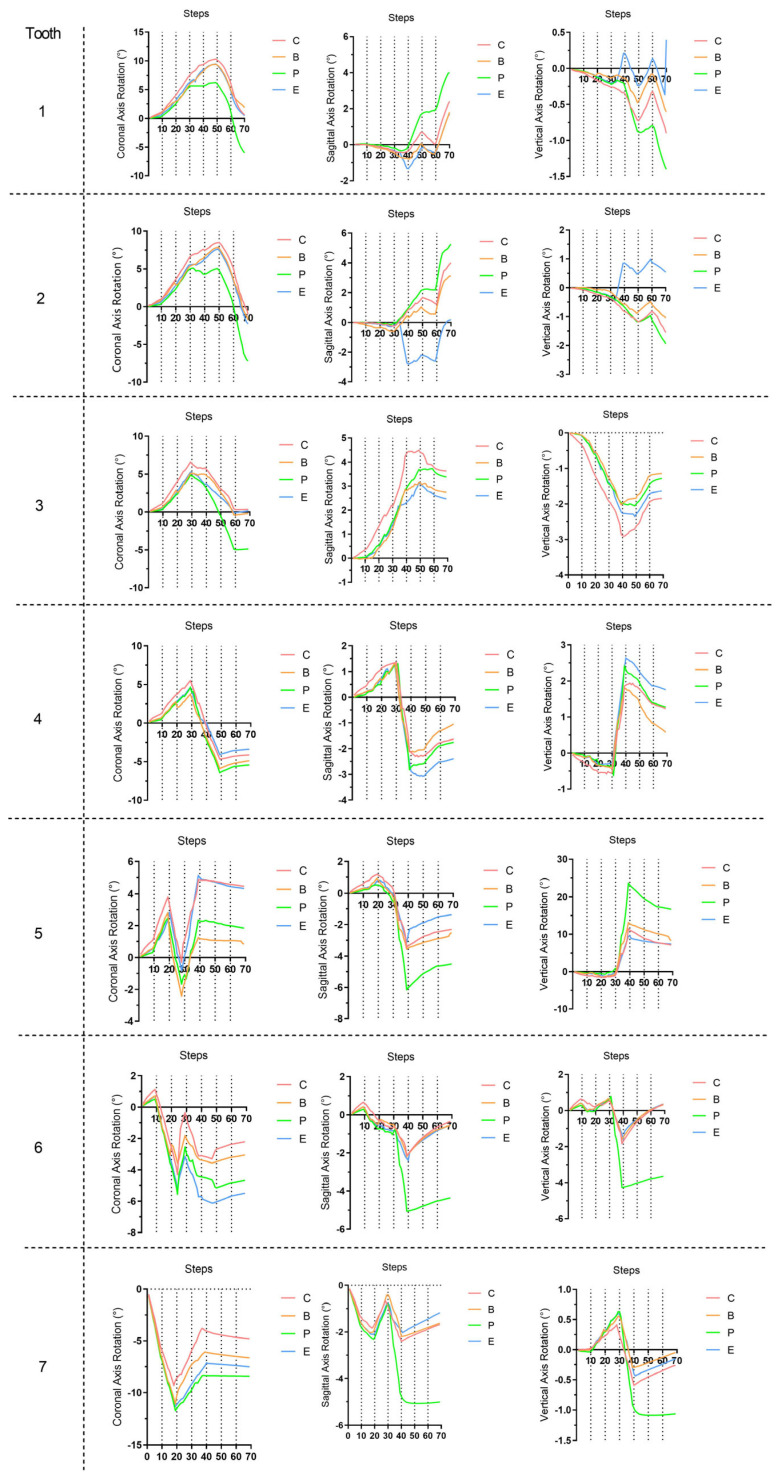
Three-dimensional rotation of the long axis of each tooth during the 70 steps. The positive axis was established following the right-hand rule. C, control group; B, buccal temporary anchorage device (TAD) group; P, palatal TAD group; E, Class II elastic group. The tooth numbering system adheres to the FDI tooth numbering system.

**Figure 9 bioengineering-11-00003-f009:**
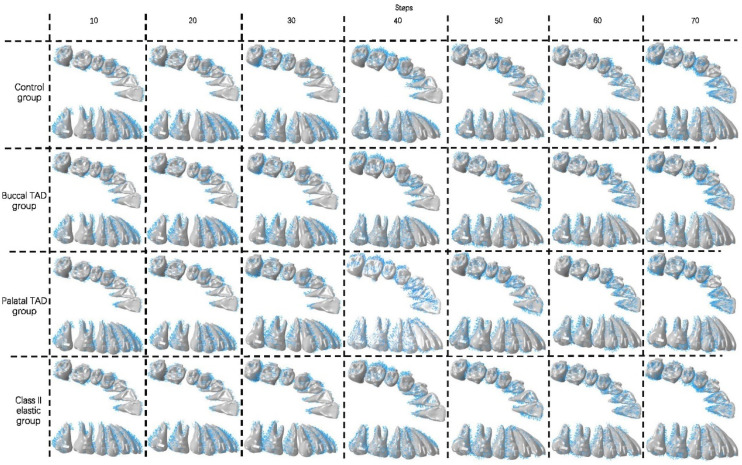
Teeth displacement tendencies (blue arrows) at different treatment staging steps of the four groups. TAD, temporary anchorage device.

**Figure 10 bioengineering-11-00003-f010:**
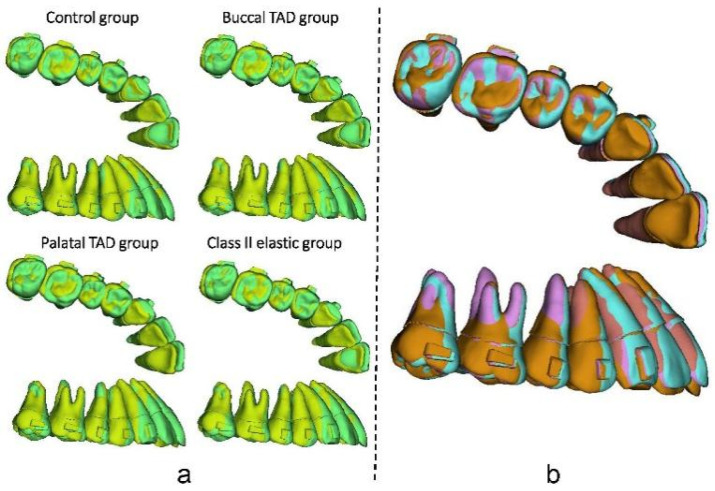
(**a**) Comparation of the initial dentition (green) and the dentition at the 70th step (yellow) of the four groups. (**b**) The final dentition at the 70th step of the control group (blue), buccal TAD group (red), palatal TAD group (orange), and Class II elastic group (purple). TAD, temporary anchorage device.

## Data Availability

The authors confirm that the data supporting the findings of this study are available within the article and its Appendix A.

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
