# Peer review of "Effect of Different Anchorage Reinforcement Methods on Long-Term Maxillary Whole Arch Distalization with Clear Aligner: A 4D Finite Element Study with Staging Simulation"

_bioengineering, 2023, doi:10.3390/bioengineering11010003_

Round 1

Reviewer 1 Report (Previous Reviewer 3)

Comments and Suggestions for Authors

The corrections made by the authors, considering the reviewer's suggestions, greatly improved the article. I consider it suitable for publication

Author Response

Thank you for your approval.

Reviewer 2 Report (New Reviewer)

Comments and Suggestions for Authors

The article is well written and interesting, it finds notable application today where the use of aligners is increasingly widespread. I advise the authors to implement the introduction with some more references relating to the literature already present on aligners and their ability to distalize, I suggest implementing with the following article: "Nocini R, Tacchino U, Ghislanzoni LH, Bertossi D, Ricciardi G, et al. Minerva Stomatol. 2020 Dec;69(6):329-334. doi: 10.23736/S0026-4970.20.04128-X. PMID: 33393274." The images used by the authors are clear and easy to understand.

Comments on the Quality of English Language

English language is fine

Author Response

Thank you for your approval. We have added this citation in the Introduction section.

Reviewer 3 Report (New Reviewer)

Comments and Suggestions for Authors

Dear authors, congratulations for the study and your contribution to the research. The article is well structured and previously well improved. 

The investigation made by authors is interesting and the sections proposed in the article, particularly “methods” and “discussion” are well detailed. 

 However I suggest to add a specific section to the limitations of the study and improve the conclusions , specifying in this last section which are the future perspectives and the aspects that need to be deepened.

Author Response

Thank you for your approval. We have improved the Discussion section and the Conclusion section accordingly:

Limitations of this study were also obvious, and further studies are needed. For the first time, this study proposed a feasible long-term orthodontic simulation method for CA treatment. However, only combined with further in vitro experiments and clinical trial, that the accuracy of the orthodontic tooth movement simulation method could be certified and improved. Also, in this study, elastics were all exerted directly on the CA simulating the use of precision cuts. Traction forces from lingual buttons or power arms attached to the teeth should also be investigated in further studies. Besides, all test groups have the same traction force magnitude set as 150 g. However, different amount of traction forces may have different biomechanical consequences, due to the potential offtrack of CA, which should be further investigated. Moreover, a lot of sim-plification was made in the current bone remodeling simulation method to gain a bal-ance between the computational efficiency and accuracy, which clinical accuracy should also be justified. However, we believe that this study opened a door for long-term orthodontic simulation for CA treatment. Despite the current proposed maxillary whole arch distalization scenario, with the development of FEM, more ac-curate and clinically significant results for all kinds of teeth movement design can be gained in the future.

  1. Conclusion

In this study, a time-dependent 4D FEM was developed to predict orthodontic tooth movement during maxillary dentition distalization with CA. Using this method, the morphological changes of CA during long-term orthodontic treatment were suc-cessfully simulated. The achieved results indicated:

  1. Compared with the other anchorage reinforcement measurements such as Class II elastics and buccal TADs, palatal TADs were recommended during maxillary dentition distalization with CA for better anterior teeth anchorage maintenance both sagittally and vertically.
  2. Attention should be paid to possible increasement of arch width at the molar and premolar levels.
  3. Overbite of front teeth may become shallow during the process, and compensa-tion measures should be carried out timely.

Reviewer 4 Report (New Reviewer)

Comments and Suggestions for Authors

The authors aimed to examine the impact of various anchorage designs on the long-term maxillary whole arch distalization using clear aligners, buccal temporary anchorage device (TAD) and palatal TAD, designed employing an innovative automatic staging simulation method previously introduced in previous author’s group studies.

The results indicated that palatal TAD group had the largest retraction of incisors, followed by the buccal TAD group and the Class II elastic group, and the least in control group.

The results and conclusions are based on interesting data and help to extend the knowledge on the topic in the future.

Thus, I have no further comments against the manuscript.

Author Response

Thank you for your approval. 

This manuscript is a resubmission of an earlier submission. The following is a list of the peer review reports and author responses from that submission.

Round 1

Reviewer 1 Report

Comments and Suggestions for Authors

Please define the magnitude of the force and its effect on the result of this study ( Higher forces can dislodge the aligner)

Please define the type of tooth movement and couple /force ratio for different teeth. For example, it is well known that bodily movement is very difficult with aligners, and here it is not clear if we discuss distalization as bodily movement, controlled tipping, or uncontrolled tipping.

Please analyze the magnitude of the deformation of plastic that causes the aligner not to sit correctly during movement and the effect on the force system of aligners in all three dimensions.

Please discuss the limitations of the design in much more detail so the readers do not make a premature conclusion.

Reviewer 2 Report

Comments and Suggestions for Authors

Palatal TADS are usually those put in the midpalatal area; however, in the materials and methods, authors have specified that those they called palatal TADS are interradicular. Why did you put the implants in the interradicular sites, and not in the midpalatal area? Midpalatal TADS are currently one of the most used sites of anchorage, so I think that another group should be added.

For your Finite element analysis, what type of CA have you considered? What material and what thickness? And why did you choose that

Comments on the Quality of English Language

There are some English mistakes that should be fixed (i.e. "and different anchorage designs was investigated")

Please leave a space, after full stops.

Reviewer 3 Report

Comments and Suggestions for Authors The topic is very interesting and current. The article is well structured in accordance with the journal's standards and the theme appropriate to the journal's scope. The introduction addresses the topic objectively and succinctly. The materials and methods are well described and the results are clearly presented. The discussion and conclusions are appropriate to the results. References are adequate and current. The authors should only rectify some gaps:
  • Line 3 - "whole" is written in a different font. It must be corrected.
  • The discussion should address future perspectives, with a view to overcoming the aforementioned limitations.
  After these small corrections, I recommend publishing the article.